# Intermittent aerobic-resistance interval training versus continues aerobic training: Improvement in cardiac electrophysiologic and anthropometric measures in male patients post myocadiac infarction, a randomized control trial

Horesh Dor-Haim[1], Michal Horowitz[2], Eldad Yaakobi[3], Sara Katzburg[1,4], Sharon Barak[5,6]*

1 O2 Health Promotion and Sports Medicine Department, Givat Ram, Jerusalem, Israel, 2 The laboratory of Environmental Physiology Department of Physiology, Faculty of Dentistry Hebrew University of Jerusalem, Hadassah Ein Kerem Campus Jerusalem, Jerusalem, Israel, 3 The Sagol Center for Hyperbaric Medicine and Research, Yitzhak Shamir Medical Center, Be'er Ya'akov, Israel, 4 Department of Developmental Biology and Cancer Research, Israel-Canada Medical Research Institute, Faculty of Medicine, Hebrew University of Jerusalem, Jerusalem, Israel, 5 Department of Pediatric Rehabilitation, The Edmond and Lily Safra Children's Hospital, The Chaim Sheba Medical Center, Ramat Gan, Israel, 6 Kaye Academic College of Education, Beer-Sheba, Israel

* sharoni.baraki@gmail.com

## Abstract

### Purpose

Exercise is a valuable intervention modality for patients post-myocardial infarction (MI). Aerobic and resistance training are both commonly used separately in cardiac rehabilitation. However, the effect of aerobic interval exercise combined with alternating sets of resistance training (super-circuit training, SCT) on cardiac electrophysiologic and anthropometric measures had not been thoroughly investigated.

### Aim

The primary objective of this study was to compare the effectiveness of moderate-intensity continuous-aerobic training (CAT) vs. SCT on cardiac electrical measures (resting electro-cardiographic, ECG; a nd heart rate variability, HRV) in patients' post-MI presenting reduced left ventricular function. Second, to examine its effect on anthropometric measures.

### Material and methods

Twenty-nine men post-MI with reduced left ventricular function were assigned randomly to either 12 weeks of CAT (n = 15) or SCT (n = 14). CAT group performed moderate-intensity activity. SCT group performed high-intensity exercise, alternating between resistance and

**Data Availability Statement:** All relevant data are within the paper and its Supporting information files.

**Funding:** The author(s) received no specific funding for this work.

**Competing interests:** The authors have declared that no competing interests exist.

aerobic training. Differences between CAT and SCT groups were done using independent t-tests, paired t-tests and effect size (ES).

## Results

Participants in both groups improved their HRV measures (increase in HFnu; $p < 0.05$; ES > 0.51) and ECG (reduction in QT-dispersion; $p < 0.05$; ES > 0.51). Only the SCT group had significant improvements in waist circumference ($p < 0.05$).

## Conclusion

Exercise improves cardiac electrical measures post-MI. However, in comparison to CAT, SCT may yield greater anthropometric changes. In order to have improvements in cardiac electrical stability, clinicians working with post-MI patients may use both CAT and SCT. However, SCT might result in greater improvements.

## Introduction

Exercise training is considered an essential approach component for rehabilitation and secondary prevention of coronary heart disease [1] and usually consists of moderately intense continuous aerobic training (CAT) [2]. The benefits of CAT in cardiac patients are well established [2–4]. However, CAT leads to only a minor increase in muscle mass or strength and is usually associated with a more pronounced improvement in aerobic exercise capacity [5]. Thus, resistance training is part of every guideline for exercise-based cardiac rehabilitation [1]. There is evidence for the safety and efficacy of resistance training in cardiac patients [6, 7]. For example, studies showed that appropriate resistance training induces metabolic, histochemical, and functional adaptations in skeletal muscles. Resistance training also increases muscle mass and strength effectively [8].

In cardiac patients, high-intensity interval training is safe, feasible, and is more effective than CAT in several outcome measures (e.g., peak oxygen consumption) [9, 10]. Super-circuit training (SCT) is a novel type of training that involves two combined training modalities: resistance-training set simultaneously followed by an aerobic exercise interval. Studies on SCT showed that this type of training in healthy individuals increased strength and aerobic capacity. Compared to CAT, intensive SCT among healthy overweight middle-aged men led to a significant decrease in metabolic indices [8]. These results are of special interest in cardiac rehabilitation as studies have shown that obesity can increase the risk of sudden death due to arrhythmic disorders [11] and changes in the autonomic nervous system [12–14]. In a previous study, we compared the effects of CAT to SCT on men post-myocardial infarction (MI) with reduced left ventricle function. SCT yielded better benefits to the patient's mechanical cardiac function (left ventricle mechanical function). Moreover, compared to CAT, SCT yielded a better benefit to the patient's fitness level, namely, aerobic capacity and strength [15]. However, the effects of CAT and SCT on cardiac electrical outcome measures [resting electrocardiograph (ECG) and heart rate variability (HRV)], were not analyzed. These non-invasive measures are of special interest as they correlate with patients' cardiovascular health and electrophysiological stability in individuals with cardiac conditions (e.g., left ventricle dysfunction) [16, 17]. Since obesity is associated with adverse cardiac events [11], it was also vital to examine the CAT and SCT effects on anthropometric measures.

The primary objective of the study was to compare the effectiveness of SCT vs. CAT on cardiac electrical outcome measures (i.e., resting ECG and HRV) in post-MI patients with reduced left ventricle function. Our second objective was to compare the effects of these two regimes on patient's anthropometric measures (body mass index, BMI; and waist circumference). We hypothesized that both CAT and SCT would be beneficial to improve electrophysiological measures. However, SCT may be more effective to improve cardiac ECG and anthropometric measures.

## Materials and methods

### Study participants

Post-MI male patients were referred to the cardiac rehabilitation center at Hadassah Mt. Scopus 6±10 weeks' post-hospitalization due to acute MI. Inclusion criteria: 1) echo testing exhibited reduced left ventricle function (ejection fraction < 45%). Ejection fraction of < 45% was selected as it is one of the predictors of poor prognosis and increased mortality in hospitalized patients [18]; and 3) New York heart association level 3 or less. New York heart association was used as an inclusion criteria as it commonly used as a fundamental tool for risk stratification of heart failure and determines clinical trial eligibility and candidacy for drugs and devices [19]. and 3) patients were able to attend regularly a supervised exercise program.

Exclusion criteria: 1) chronic atrial fibrillation, 2) severe valvular disease, 3) angina or peripheral arterial occlusive disease, and 4) cerebrovascular or musculoskeletal disease-preventing exercise testing or training. The study was approved by the Helsinki ethics committee, Hadassah medical center (0440-12-HMO, ClinicalTrials.gov Identifier: NCT01912690). All participants gave written informed consent.

### Study procedures

Participants were randomly assigned (sealed envelope method) to either twice a week CAT or SCT by cardiac rehabilitation staff who were not involved in the research study. The technical work and analysis of the results were blinded. The care providers were not blinded. During the trial, drug therapy remained unchanged, and patients with type 2 diabetes, or hypertensive, were not regulated in their drug therapy dosage during the 12-week intervention. A participant was deducted from the study if he developed adverse effects such as chest tightness, ECG changes, or severe arrhythmias.

Patients in both groups started each training session with five minutes warm-up, followed by either CAT or SCT. Throughout the training sessions, exercise intensity was defined using heart rate measurement (Polar Electro, Kempele, Finland; or Nihon Kohden ECG telemetry) and calculation of heart rate reserve (i.e., maximal heart rate—resting heart rate). Participants' maximal heart rate was established using a baseline Bruce graded exercise tolerance test (GE Marquette CASE 8000 Exercising Testing System). The graded exercise tolerance test was terminated if the patient presented a > 10 mmHg decrease in systolic blood pressure with increasing workload, a moderate-to-severe angina, evidence of significant arrhythmia's (e.g., > 3 premature ventricular contractions in a row), unusual or severe shortness of breath, evidence of poor perfusion, equipment's mal function, or if the patient requested to stop the test. At the end of the exercise, both groups conducted five minutes of gradual cool down. For additional information on the test's protocol, refer to Dor-Haim et al. [15].

CAT group participants exercised continually at 60% to 70% of their heart rate reserve using a treadmill (TechnoGym) and lower and upper extremity cycle ergometer (Star Terk). A modified Borg 1-to-10 scale was used to assess the rate of perceived exertion, during and after each training session. Speed and inclination of the treadmill or resistance and cadence of the

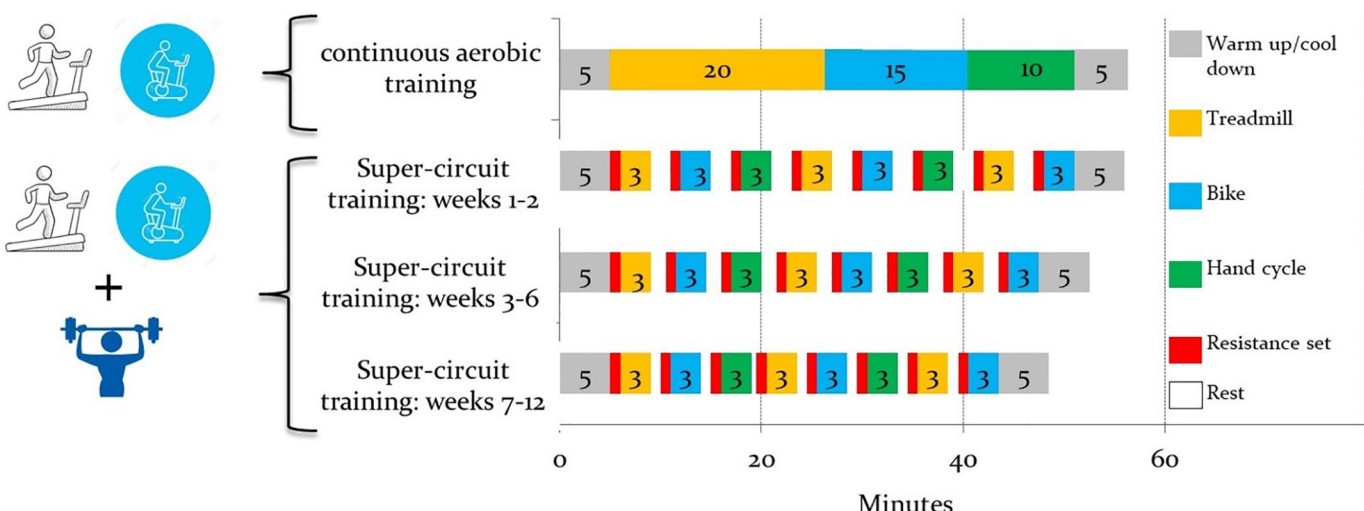

**Fig 1. Super-circuit training and continuous-aerobic training—Training program (adapted from Dor-Haim et al., 2018) [15].** Continues aerobic training group conducted only continues aerobic training and super-circuit training group conducted both aerobic and strength training.

cycle ergometer were adjusted continuously, to ensure that every training session was carried out at the assigned heart rate reserve throughout the training period (Fig 1).

The SCT group performed high-intensity exercise, alternating between resistance and aerobic training. One resistant training set was followed by three minutes of aerobic exercises and a resting period, repeated eight times. Each exercise consisted of one set of 15 repetitions on a Cybex resistance training machine. In the first two weeks of the program, the training intensity was light, RM 1 was assessed in the second day, according to the method of Kraemer and Fry [20]. Starting from- 30% of 1-repetition maximum, and progressively increased to 50% of 1-repetition maximum. Participants had to maintain appropriate lifting techniques, not hold their breath to avoid the Valsalva maneuver, and carefully alternate positions to adapt to the blood pressure orthostatic changes. The aerobic intensity was determined at 75% to 85% of heart rate reserve. Resting periods between intervals was minimal, and gradually decreased from two minutes in the first two weeks to 1.5 minutes in weeks 3–6 and 1 minute in weeks 7–12 (Fig 1).

For a more detailed description of the study design and exercise protocol, refer to Dor-Haim et al (2018) [15].

## Outcome measures

All outcome measures were evaluated at baseline and post-12 weeks of the exercise training.

**Cardiac electrical measures—Primary outcome measures.**

1. Resting electrocardiogram (ECG)–Twelve lead resting ECG for a duration of 10 minutes was recorded before the graded exercise tolerance test (Cardio scape PC ECG, version 3.1) using a fixed speed of 25 mm/sec and standardization of 1 mm as 1 millivolt throughout the measurement. QT interval was measured from the QRS complex onset point to the T wave offset point (the return of T wave to baseline). QT-dispersion (QTd) and Corrected QTd (QTcd) was calculated as the difference between the maximum and minimum QT intervals for any of the 12 leads [21].

2. Heart rate variability (HRV): Measurements of heart rate (Cardio scape PC ECG, version 3.1) were performed at rest. For analysis and calculation of HRV, the Kubios HRV software 2.0 was used [22]. The data were visually examined for quality control. Periods of excessive noise were removed from the analysis. Spectral analysis was used to calculate high-frequency (0.15–0.40 Hz) and low frequency (0.04–0.15 Hz) normalized units (HFnu and LFnu, respectively) using Fast Fourier Transformation. The ratio of low-to-high frequency (LF/HF) band powers was evaluated. A frequency-domain analysis method was selected for comparison to time-domain measures, as frequency-based analyses provide a more accurate interpretation of parasympathetic and sympathetic influences [16, 23].

**Anthropometric measures.** Two anthropometric parameters were measured: body mass index (BMI; weight in kg/height in $m^2$) and waist circumference (measured 2.5 centimeters above the umbilicus) [24]. All anthropometric measures were conducted by one technician.

**Data analysis.** Normality assumption was evaluated using the Shapiro-Wilk test [25]. The analysis revealed the all study variables are normally distributed ($p > 0.05$) with $W$ ranging from 0.90 to 0.94. In addition, in order to test whether or not data are missing completely at random (i.e., $p > 0.05$), Little's test of missing completely at random test [26] was conducted. This test is useful for testing the assumption of missing completely at random for multivariate, partially observed quantitative data [27]. The test's assumption is that the missingness of the data is independent of both the unobserved and the observed data [28]. In the current study, Little's Missing data analysis showed that data were missing completely at random (Chi-square distance = 89.50, p = 0.32). This information was added to the statistical analysis section.

Differences between CAT and SCT groups in the various outcome measures at both pre and post-tests were examined using independent t-tests. To evaluate within-group changes from pre-to-post-test in the various outcome measures, paired t-tests were conducted (alpha level p<0.05, adjusted to 0.025 using the Bonferroni procedure). For HRV measurements, alpha level was adjusted to 0.016 (0.05/3 = 0.016). Effect sizes (ES) using Cohen's $d$ [29] were also calculated in order to quantify the degree of change in each study group. A correction for the dependence among means was done using the correlations between the two means following Morris and DeShon's equation [30]. In general, values smaller and equal to 0.20 were considered trivial ES, values between 0.21 and 0.50 as small ES, values 0.51–0.80 as moderate ES, and values greater than 0.80 as large ES [29].

A posthoc power analysis was conducted using the main outcome measures, mainly, ECG and HRV. The average ES of these variables in the CAT and SCT groups were 1.78 and 0.78, respectively. Using posthoc power analysis for the ESs, $\alpha = 0.05$, and the study's sample size, the power to detect differences between two dependent variables (matched pairs) was 0.90 and 0.80, respectively. The mean ES difference between the groups was 0.78. Given this ES, $\alpha = 0.05$, and the study's sample size, the study's power to detect between-group differences (independent t-test) was 0.78. Power analysis calculations were done using G*Power (version 3.0.10).

## Results

### Study participants

Fifty-eight patients were referred to the cardiac rehabilitation center. Ten participants declined to participate or did not meet the inclusion criteria. Forty-eight participants (mean age = 59.14 years old + 8.92; range: 42.00–75.00) were assigned randomly to the CAT (n = 26) or SCT (n = 22). No statistically significant differences between CAT and SCT groups in age were

**Table 1. Participant's demographic and clinical background.**

| Demographic/ clinical measures | group | Continues aerobic training | Super-circuit training | Between-group differences: t-score OR Chi-squre (p value) |
|---|---|---|---|---|
| | | Mean (SD) OR n (%) | Mean (SD) OR n (%) | |
| Age, tears: mean (SD) | | 61.21 (8.03) | 57.07 (9.57) | -1.24 (0.22) |
| Pharmacological treatment: n (%) | Beta-blockers | 17 (65.3) | 12 (57.1) | 0.33 (0.56) |
| | An angiotensin-converting-enzyme inhibitor | 21 (80.7) | 15 (68.1) | 0.98 (0.32) |
| | Diuretics | 6 (23.0) | 6 (27.2) | 0.11 (0.74) |
| | Statins | 19 (73.0) | 18 (81.8) | 0.41 (0.51) |
| | Anti-coagulations | 8 (30.7) | 7 (31.8) | 0.00 (0.94) |
| Co-morbidities | Diabetes mellitus | 8 (30.7) | 4 (18.1) | 0.90 (0.34) |
| | Hypertension | 13 (50.0) | 12 (54.5) | 0.07 (0.78) |
| | Obesity | 6 (23.0) | 4 (18.1) | 0.17 (0.67) |

Notes: SD, standard deviation

observed (mean age = 61.21 ± 8.03 vs. 57.07 ± 9.57, respectively; t statistic = -1.24; p = 0.22; Table 1).

In the CAT group, post-test data were available for 15 participants. In the SCT group, post-test data were available for 14 participants. In both CAT and SCT groups most participants were treated with beta-blockers (65.3 and 54.5%, respectively), an angiotensin converting enzyme inhibitor (80.7 and 68.1%, respectively), and statins (73.0 and 81.8%, respectively). In addition, in both CAT and SCT, most participants' had hypertension (50.0 and 54.5%, respectively; Table 1). For study participants flow-chart, refer to Fig 2.

## Cardiac electrical measures

The CAT group showed significant differences from pre to post-test in two HRV measures (increased HFnu and decreased LFnu; $p < 0.016$, Fig 3a) and in QT ECG measures (i.e., decreased QTd and QTcd; $p < 0.025$, Fig 3b). The SCT group yielded a significant increase in HFnu ($p < 0.016$, Fig 3a). The SCT group did not demonstrate significant changes in LFnu (Fig 3a). The SCT group showed significant changes in the two ECG measures ($p < 0.025$, Fig 3b). Regarding ES, the changes observed in both study groups were moderate-to-large (Cohen's d > 0.51, Table 2). No significant between-group changes were found in the cardiac electrical measures, except for LFnu that in comparison to the SCT group, was significantly lower in the CAT group at pre-test ($p = 0.006$, Fig 3a and 3b).

## Anthropometric measures

No statistical significant between-group differences in BMI were observed in at both pre and post-tests. Similarly, in both groups participants did not present a statistically significant change from pre-to-post-test in BMI. Waist circumference of SCT groups was statistically significantly lower than this of CAT group in both pre-and-posttest. Moreover, only the SCT group presented a significant reeducation in waist circumference from pre-to-posttest ($p < 0.025$; ES = -0.55). For additional information, refer to Tables 2 and 3.

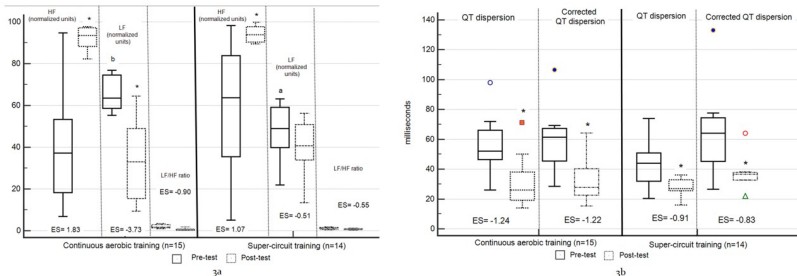

Met inclusion criteria, n=48

Super-circuit training, n=22 — Attrition, n=2

Random assignments

Continues aerobic training, n=26 — Attrition, n=1

Started pre-tests, n=20

Started pre-tests, n=25

Started training, n=20 — Attrition, n=4

Started training, n=25 — Attrition, n=6

Completed the program but did not partake in post-tests, n=2

Conducted post-tests, n=14

Conducted post-test, n=15

Completed the program but did not partake in post-tests, n=4

**Fig 2. Study participants' flow chart (adapted from Dor-Haim et al., 2018) [15].** Forty-eight participants were assigned randomly to the continues aerobic training (n = 26) or super circuit training (n = 22). In the continues aerobic training group, 19 participants completed the program. Post-test data were available for 15 participants. In the super circuit training group, 16 participants completed the program, post-test data were available for 14 participants.

**Fig 3. Within and between-group differences cardiac electrical measures. 3a—heart rate variability. 3b—electrocardiograph**—The central box represents the values from the lower to upper quartile (25 to 75 percentile). The vertical line extends from minimum to maximum values, excluding the outside values displayed as separate points. An outside value is defined as a value that is smaller than the lower quartile minus 1.5 times the interquartile range, or larger than the upper quartile plus 1.5 times the interquartile range; the middle line represents the median; *significant within-group changes from pre to post-test; [a]between-group differences—significantly different than CAT group; [b]between-group differences—significantly different than SCT group.

**Table 2. Continuous aerobic training and super-circuit training effect sizes.**

| Primary and secondary outcome measures | | Cohen's *d* effect size | |
|---|---|---|---|
| | Training group | Continues aerobic training | Super-circuit training |
| Resting electrocardiograph | QT dispersion | -1.24 | 0.91 |
| | Corrected QT dispersion | -1.22 | -0.83 |
| Heart rate variability | $HF_{nu}$ | 1.839 | 1.07 |
| | $LF_{nu}$ | -3.73 | -0.51 |
| | LF/HF | -0.90 | -0.55 |
| Anthropometric measures | Body mass index | -0.08 | -0.34 |
| | Waist circumference (cm) | -0.06 | -0.55 |

Note: Cohen's d calculation: mean Δ/standard deviation average from two means. Dark gray cells -moderate and large differences (Cohen's $d \geq 0.51$). Light gray cells—small differences (Cohen's $d = 0.21$–$0.50$). White cells—trivial differences (Cohen's d $\leq 0.20$).

*Cohen's *d* is based on a single pooled standard deviation. Cohen's *d* was corrected for dependence between means, using Morris and DeShon's equation [30].

**Table 3. Anthropometric measures: Within and between group differences.**

| Variables | | Continues aerobic training | | | Super-circuit training | | | Between-groups analysis | |
|---|---|---|---|---|---|---|---|---|---|
| | | Pre-test: mean (SD) | Post-test: mean (SD) | Statistic t (p value) | Pre-test: mean (SD) | Post-test: mean (SD) | Statistic t (p value) | Pre-test: statistic t (p value) | Post-test: statistic t (p value) |
| Anthropometric Measures | Body mass index | 29.35 (3.81) | 29.04 (3.65) | -1.488 (0.164) | 28.14 (2.90) | 27.14 (2.73) | -2.72 (0.026) | -0.789 (0.439) | -1.305 (0.207) |
| | Waist circumference (cm) | 107.56 (7.36) | 107.12 (7.88) | -1.01 (0.302) | 98.66 (7.17) | 94.66 (6.94) | -6.35 (0.0002) * | -2.59 (0.019) | -3.55 (0.002) * |

Notes: SD, standard deviation;

* significant within or between-group differences at the $p < 0.05$ (alpha level of body composition measurements was adjusted to 0.025 using the Bonferroni procedure.

## Discussion

This study's primary finding was demonstrating that -exercise, primarily SCT post-MI in patients with reduced left ventricular function was valuable to improve electrocardiographic measures (resting ECG and HRV). Secondly, only SCT was effective to reduce waist circumference. The study findings are consistent with the hypothesis that SCT may be more effective for patients post MI with reduced left ventricular function. In our previous study [15], SCT was more efficient than CAT in enhancing cardiac mechanical systolic (ejection fraction) and diastolic function (mitral inflow E velocity to tissue Doppler E/e' ratio). Resting ECG measure such as QTd and HRV are key electrophysiological markers post-MI in ischemic patients. These measures are related to electrical remodeling of the heart post-MI, arrhythmias such as atrial fibrillation [31], while such modeling in the ventricle may cause potentially lethal ventricular arrhythmias [32]. Therefore, the effects of CAT and SCT on electrocardiograph measures were essential for investigation.

The results presented here indicate an improvement in both QTd and HRV by both CAT and SCT regimes. More specifically, in both groups, QTd and QTdc decreased (p<0.01). Moreover, both groups showed a significant increase in HF spectral component, attributed to an increase in parasympathetic tone and antiarrhythmic protection [33]. Our results are consistent with other studies indicating the benefits of exercise training for the heart's electrical

stability post-MI [17]. The beneficial effects of both CAT and SCT on ECG and HRV measures are encouraging as such improvements are associated with reduced cardiovascular disease burden and mortality. Other clinical and animal studies showed that QTd and HRV are independent non-invasive markers for ventricular arrhythmia and sudden cardiac death following MI [34, 35]. QTd also serves as a marker for electrical inhomogeneity during myocardial repolarization. In addition, HRV is related to autonomic nervous system regulation. In MI survivors, lower HRV was associated with the remodeling of the autonomic nervous system [36] and an increased risk of tachyarrhythmia [37]. Therefore, improved HRV and QTd in the current study by safe exercise protocol may elicit favorable cardiac electrical reversed remodeling among patients' post-MI with reduced left ventricle function.

Few studies showed a deleterious arrhythmogenic effect of exercise, mostly in vigorous exercise [38, 39]. In the current study, we demonstrated that both exercise regimes resulted in an improvement in electrical markers. However, the SCT method resulted in a better cardiac intrinsic recovery, and thus, may yield a better prognosis post-MI. Animal studies also demonstrated that increased exercise intensity improves cardiovascular electrical stability in a dose-response relationship [40]. In the current study, the only difference between the two training regimes was found in the LF component, which was decreased in the CAT group but not in the SCT group. The LF component represents the interaction of the sympathetic and parasympathetic nervous systems. Exercise training induces adaptations in HRV outcomes with a shift of autonomic balance toward higher parasympathetic activity, consistent with improved cardiac health [16]. The decrease in the LF component is commonly observed in MI survivors and is attributed to increased sympathetic tone and increased risk of sudden death [41]. Hence, from the electrophysiological aspect attenuated LF reduction seen only in the SCT group may represent a relatively improved clinical reaction.

Both training groups did not show significant changes in pre-post BMI, suggesting that participants did not lose or gain weight. However, the SCT group presented significant reductions in waist circumference. These results showed that only the SCT group decreased their abdominal visceral fat [42, 43], which is one component of metabolic syndrome [44]. Reduction of body fat without losing weight may indicate a gain in muscle mass at the expense of fat mass loss [45]. In order to better understand programs' effect on body composition, additional anthropometric measures should be included in future studies. Compared to subjects with high muscle/low fat, the risk of arrhythmia due to cardiac intrinsic electrical instability is significantly higher in people with a high fat /low muscle ratio [46]. It is also important to note that abnormal autonomic regulation is prevalent in patients with metabolic disorder. For example, several HRV studies showed abnormalities in autonomic nervous control in obese and overweight subjects [47, 48]. In overweight individuals, a sympathovagal imbalance due to increased sympathetic activity and its association with visceral fat was observed [49]. In another 6-month study, aerobics alone was compared to a combined aerobic and resistance training. Both regimes significantly decreased abdominal visceral fat, but combined aerobic and resistance training was more effective [43].

The current study was subject to several limitations. First, the generalizability of the results is in question owing to the single center and the small sample size of male only patients in each study group. The inclusion of only male patients is a common characteristic of numerous studies examining exercise training effects on heart failure patients. For example, in a recent meta-analysis in the subject it was reported that the majority of patients in cardiac rehabilitation and exercise training trials are males (77%) [50]. Second, anthropometric measures were limited in this study and could be expended to measure more variables such as body composition and blood chemistry. Last, the study period of three months is too short to draw conclusions in regarding to prognostic indicators for the two exercise groups, such as re-hospitalizations and mortality.

## Conclusions

In conclusion, the current study showed that electrocardiographic measures post-MI stand to benefit from both training regimes, namely, CAT and SCT. Nevertheless, in comparison to aerobic training alone (i.e., CAT), SCT may yield better benefits to autonomic balance and anthropometric measures. Considering the effect of exercise on MI patients, it is vital to introduce novel training modalities that may enhance autonomic balance, intrinsic mycardial recovery and health related anthropometric factors post-MI.

## Supporting information

**S1 Data.**
(XLSX)

## Author Contributions

**Conceptualization:** Horesh Dor-Haim, Michal Horowitz, Eldad Yaakobi, Sara Katzburg.

**Data curation:** Horesh Dor-Haim, Sara Katzburg.

**Formal analysis:** Sharon Barak.

**Investigation:** Horesh Dor-Haim, Michal Horowitz, Eldad Yaakobi.

**Methodology:** Horesh Dor-Haim, Michal Horowitz, Eldad Yaakobi, Sharon Barak.

**Project administration:** Horesh Dor-Haim.

**Resources:** Horesh Dor-Haim.

**Supervision:** Horesh Dor-Haim, Eldad Yaakobi.

**Writing – original draft:** Horesh Dor-Haim, Sara Katzburg, Sharon Barak.

**Writing – review & editing:** Michal Horowitz, Eldad Yaakobi, Sharon Barak.

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
