## [Decision Letter · Decision Letter 0]

23 Feb 2022

PONE-D-22-01160Intermittent Aerobic-Resistance Interval Training: improvement in cardiac electrophysiologic and cardio-metabolic measures, a randomized control trialPLOS ONE

Dear Dr. Barak,

Thank you for submitting your manuscript to PLOS ONE. After careful consideration, we feel that it has merit but does not fully meet PLOS ONE’s publication criteria as it currently stands. Therefore, we invite you to submit a revised version of the manuscript that addresses the points raised during the review process.

We look forward to receiving your revised manuscript.

Kind regards,

Walid Kamal Abdelbasset, Ph.D.

Academic Editor

PLOS ONE

Journal Requirements:

Reviewers' comments:

Reviewer's Responses to Questions

**Comments to the Author**

1. Is the manuscript technically sound, and do the data support the conclusions?

Reviewer #1: Yes

Reviewer #2: Partly

Reviewer #3: Partly

2. Has the statistical analysis been performed appropriately and rigorously? 

Reviewer #1: Yes

Reviewer #2: No

Reviewer #3: Yes

3. Have the authors made all data underlying the findings in their manuscript fully available?

Reviewer #1: Yes

Reviewer #2: No

Reviewer #3: No

4. Is the manuscript presented in an intelligible fashion and written in standard English?

Reviewer #1: Yes

Reviewer #2: Yes

Reviewer #3: Yes

5. Review Comments to the Author

Reviewer #1: The title should be rewritten to express the real comparative study work.. How would you express that all participants are of same gender ?..The author didn't mention minium ejection fraction for inclusion criteria to provide patients safety

Reviewer #2: Interesting paper, but there are things need to be revised and considered before publication

please see the attached file for recommendations within the manuscript as comments.

A major part is:

- There is no description to the graded exercise test used in the study in the methods section

- There is no demographical data for the participants

- In the results tables, add a table for HRV and QT before and after each training program.

Reviewer #3: PONE-D-22-01160: statistical review

SUMMARY. This study compares the effectiveness of super-circuit training (SCT) versus continuous-aerobic training (CAT) on cardiac outcomes and secondary biometrical outcomes in post-MI males with reduced left ventricle function. It is a follow-up of a previous study, already published in PLOSONE. The statistical analysis correctly relies on ANOVA methods and results seem sound. There are however three points that should be clarified: see the major issues below. I also append some specific points that should be addressed.

MAJOR ISSUES

1. Age ranges from 42 to 75 years and can be a confounder. Please test age differences between the CAT and the SCT group (a standard T-test will do the job). Two are the possible outcomes. Either age means do not significantly differ between groups; in this case the randomization procedure did a good job and the authors just need to display the result of the t-test, to reassure the reader. Or one of the two groups is significantly older (or younger) than the other one; in this case the ANOVA analysis must be extended to a regression analysis, where age is included as a confounder.

2. Page 9, Study participants section: "Fifty-eight patients ... 14 participants". As it often happens, there are missing values and drop-outs in this study. This is not necessarily a problem, provided that the missing values are ignorable. Missing values are ignorable when the probability of a missing value does not depend on the unobserved value. Do we have any information about the missing outcomes? If we can assume that the missing values are missing at random (i.e., neither older nor younger than the included subjects; neither in better nor in worse health status, and so on), then we can safely ignore them, as the authors did in their analysis. Otherwise, the results of the analysis could be possibly questionable.

3. Normality. The whole analysis relies on an assumption of normally-distributed outcomes. Could the authors give some evidence of this assumption? If serious departures from normality are detected, then p-values could be biased.

SPECIFIC POINTS

1. Abstract, first line: "Exercise is a valuable intervention modality post-myocardial infarction (MI)." I think a "in" or a "for" is missing here: please check.

2. An obvious limitation of the study is that women are not considered. This should be remarked in the discussion, perhaps referring to other studies that include women.

3. Figure 3: "The vertical line extends from minimum to maximum values, excluding the outside values displayed as separate points". Please clarify how were outliers defined, as there is not a universal definition of outlying observations.

4. Data availability. Honestly, I don't see specific reasons that would prevent the full availability of the data.

6. PLOS authors have the option to publish the peer review history of their article (what does this mean?). If published, this will include your full peer review and any attached files.

Reviewer #1: No

Reviewer #2: No

Reviewer #3: No

---

## [Author Response · Author response to Decision Letter 0]

24 Mar 2022

Reviewer 1

Thank you very much for the detailed and informative review. We hope we’ve addressed all concerns to your full satisfaction. Below we describe in detail the corrections we’ve made in the manuscript. 

1. The title should be rewritten to express the real comparative study work.

Thank you for the comment – the title was changed to represent more precisely the comparative study: Intermittent aerobic-resistance interval training versus continues aerobic training: improvement in cardiac electrophysiologic and cardio-metabolic measures in male patients post myocadiac infarction, a randomize control study

2. How would you express that all participants are of same gender?

Thank you for the comment – the title was changed to represent more precisely the male patients in study.

3. The author didn't mention minium ejection fraction for inclusion criteria to provide patients safety

Patients minimum EF was not an inclusion criterion, however minimum functional capacity was. Meaning NYHA level had to be NYHA 3 or less. Functional capacity is a well-established criterion to classify patient risk level – please note:

Caraballo, C., Desai, N. R., Mulder, H., Alhanti, B., Wilson, F. P., Fiuzat, M., ... & Ahmad, T. (2019). Clinical implications of the New York heart association classification. Journal of the American Heart Association, 8(23), e014240.‏ 

This information was added to "study participant's section. 

 

Reviewer 2

Thank you very much for the detailed and informative review. We hope we’ve addressed all concerns to your full satisfaction. Below we describe in detail the corrections we’ve made in the manuscript. 

1. A major part is: There is no description to the graded exercise test used in the study in the methods section

The graded exercise protocol was described in our previous manuscript by Dor-Haim et al., 2018. A reference to the manuscript was added to the methods section. In short, aerobic fitness was assessed by using Bruce graded exercise tolerance treadmill protocol (GE Marquette CASE 8000 Exercise Testing System). The graded exercise tolerance test was terminated if the patient presented a > 10 mmHg decrease in systolic blood pressure with increasing workload, a moderate-to-severe angina, evidence of significant arrhythmia’s (e.g., > 3 premature ventricular contractions in a row), unusual or severe shortness of breath, evidence of poor perfusion, equipment's mal function, or if the patient requested to stop the test. A short description of the protocol was also added to the methods section. 

Dor-Haim H, Barak S, Horowitz M, Yaakobi E, Katzburg S, Swissa M, Lotan C. Improvement in cardiac dysfunction with a novel circuit training method combining simultaneous aerobic-resistance exercises. A randomized trial Earnest CP, editor. PLoS ONE. 2018 January 29;13:e0188551.

2. There is no demographical data for the participants

We added the following information to the results study participants section: In both CAT and SCT groups most participants were treated with beta-blockers (66.7 and 57.1%, respectively), an angiotensin converting enzyme inhibitor (81 and 71.4%, respectively), and statins (76.7 and 85.7%, respectively). In addition, in both CAT and SCT, most participant's had hypertension (53.3 and 57.1%, respectively). "

3. In the results tables, add a table for HRV and QT before and after each training program.

We did not add the table as HRV and QT is presented as a figure (figure 3). Our apologies if that was not clear. 

3a - heart rate variability. 3b – electrocardiograph

4. Introduction – "These results are of special interest in cardiac rehabilitation as studies have shown that obesity can increase the risk of sudden death due to arrhythmic disorders [11] and changes in the autonomic system [12–14]" change to "autonomic nervous system. 

We corrected the sentence. 

5. What is the study's hypothesis? 

We hypothesized that both CAT and SCT would be beneficial to improve electrophysiological measures. However, SCT may be more effective to improve cardiac ECG and metabolic measures. The hypothesis was added to the end of the introduction.

6. Study participants description: " Inclusion criteria: 1) echo testing exhibited reduced left ventricle function (ejection fraction < 45%)". Based on which guidelines this definition of ejection fraction?

There are some predictors of poor prognosis and increased mortality in hospitalized patients, which include, among others, LVEF under 45%. This information was added to the study participants' section. 

Ouwerkerk W, Voors AA, Zwinderman AH. Factors influencing the predictive power of models for predicting mortality and/or heart failure hospitalization in patients with heart failure. JACC Heart Fail. 2014 Oct;2(5):429-36. [PubMed] 

7. Study procedures –" Participants' maximal heart rate was established using a baseline graded exercise tolerance test." What was the test? 

The graded exercise protocol was described in our previous manuscript by Dor-Haim et al., 2018. A reference to the manuscript was added to the methods section. In short, aerobic fitness was assessed by using Bruce graded exercise tolerance treadmill protocol (GE Marquette CASE 8000 Exercise Testing System). The graded exercise tolerance test was terminated if the patient presented a > 10 mmHg decrease in systolic blood pressure with increasing workload, a moderate-to-severe angina, evidence of significant arrhythmia’s (e.g., > 3 premature ventricular contractions in a row), unusual or severe shortness of breath, evidence of poor perfusion, equipment's mal function, or if the patient requested to stop the test. A short description of the protocol was also added to the methods section. 

Dor-Haim H, Barak S, Horowitz M, Yaakobi E, Katzburg S, Swissa M, Lotan C. Improvement in cardiac dysfunction with a novel circuit training method combining simultaneous aerobic-resistance exercises. A randomized trial Earnest CP, editor. PLoS ONE. 2018 January 29;13:e0188551:

8. Study procedures - "Each exercise consisted of one set of 15 repetitions on a Cybex machine." What is this machine? 

Cybex machine is a resistance training station, manufactured by Cybex company. The text was changed to clarify the content. 

9. Study procedures – "In the first two weeks of the program, the training intensity was light - 30% of 1-repetition maximum" – add information about this measurement. How and when it was made and for which muscles?

Thank you for the comment. RM 1 procedure was performed in the second day, according to the method of Kraemer and Fry, which was proved to be safe in CR. The test started from- 30% of 1-repetition maximum, and progressively increased to 50% of 1-repetition maximum. This information and the reference were added to the manuscript. 

Barnard, K. L., Adams, K. J., Swank, A. M., Mann, E., & Denny, D. M. (1999). Injuries and muscle soreness during the one repetition maximum assessment in a cardiac rehabilitation population. Journal of Cardiopulmonary Rehabilitation and Prevention, 19(1), 52-58.‏

10. Metabolic measures – "Two metabolic parameters were measured: body mass index (BMI; weight in kg/height in m2 ) and waist circumference (measured 2.5 centimeters above the umbilicus) [21]. " this was already introduced. Why repeat it?

This information was mentioned earlier only at the end of the introduction where we describe the purpose of the study. In the objective there is no description of measurement method. Therefore, we do think that we need to mention these two measures in the outcome measures section as well. In this section we provide a little information on measurement methods. 

11. Data analysis – "Differences between CAT and SCT groups in the various outcome measures at both pre and post-tests were examined using independent t-tests ". Add information about the skeweness of the variables, were all variables normally distributed or not, and in case of not-normal distribution what was the procedure? 

Normality assumption was evaluated using the Shapiro-Wilk test. The analysis revealed the all study variables are normally distributed (p > 0.05) with W ranging from 0.90 to 0.94. This information was added to statistical analysis section.

Shapiro SS, Wilk MB (1965) An analysis of variance test for normality (complete samples). Biometrika 52: 3-4.

12. Data analysis – "In general, values smaller and equal to 0.20 were considered trivial ES, values between 0.21 and 0.50 as small ES, values 0.51-0.80 as moderate ES, and values greater than 0.80 as large ES [22]." Need to write the level of significance accepted at which level.

We calculated ESs to all analyses where t-tests were also conducted. More specifically, the test was conducted in all analyses, regardless of the t test p value (greater or smaller than 0.05). We used the ES in order to understand the magnitude of the changes observed and not their statistical significance. Therefore, we do not think that we need to add significance level to the effect size. We apologize if we missed understand you. If you still think that further changes need to be conducted, we will appreciate receiving more information in the matter. 

13. Figure 3 – " Analysis level of significance was set at 0.05 and adjusted to 0.016 in heart rate variability……." This should have been mentioned earlier.

This information also appears in data analysis section. Therefore, we deleted it from Figure 3. 

14. Discussion: " This study's primary objective was to evaluate the implications of two-exercise regimes post-MI in patients with RVLF.." introduce this abbreviation. 

Because this abbreviation appears only twice, we decided to write the full name instead of using abbreviations. 

15. Discussion – normally in the first paragraph of the discussion there should be a summary of the study's findings and whether they are as expected or not. 

Thank you for the comment. The text in the first paragraph of the discussion was re-edited to demonstrate the main findings of the study.

16. Discussion – " However, the SCT group presented significant reductions in waist circumference. These results showed that only the SCT group decreased their abdominal visceral fat [35,36], which is one component of metabolic syndrome [37]." This raise a question, abdominal visceral fats are more accurately measured via bioelectrical impedance. However, using waist circumference only to measure or refer to abdominal visceral fats is subject to many confounding factors. Thus, was the measurement performed by 1 person only pre and post or was there many people involved? 

This is an important comment: Waist circumference and BMI were only a secondary objective of the study. It was measured by one technician who was very consistent in measurements. This information was added to "outcome measures section". 

Body impedance is another way to measure change in body composition, however, it tends to be inaccurate in CHF patients who tend to change their body impedance due to electrolyte, diuretics and water volume level. However, we do agree that it is one of the study limitations, therefore we added this sentence to the study limitations paragraph: Metabolic measures were limited in this study and could be expended to measure more variables such as body composition and blood chemistry.

17. Discussion: " Reduction of body fat without losing weight indicates a gain in muscle mass at the expense of fat mass loss". Unless you precisely measure it, I would advise deleting this statement. Because there are other factors which can contribute to weight other than fat mass and muscle mass. 

The text was edited: Reduction of body fat without losing weight may indicate a gain in muscle mass at the expense of fat mass loss. In order to better understand programs' effect on body composition, additional metabolic measures should be included in future studies. 

18. Discussion: " It is also important to note that abnormal autonomic regulation is prevalent in metabolic patients". Change to "patients with metabolic disorders". 

The sentence was corrected. 

19. End discussion: "Prolonged research and follow-up are required to allow firm conclusions regarding re-hospitalizations and mortality." What this statement

The sentence was rephrased: Last, the study period of three months is too short to draw conclusions regarding to the prognostic indicators for the two exercise groups such as regarding re-hospitalizations and mortality.

20. Conclusions: "….. novel training modalities that may enhance the central cardiac recovery….." Do you mean parasympathetic recovery?

Thank you, the sentence was rephrased: ….modalities that may enhance the intrinsic cardiac recovery and peripheral metabolic health-related factors post-MI.

 

Reviewer 3

Thank you very much for the detailed and informative review. We hope we’ve addressed all concerns to your full satisfaction. Below we describe in detail the corrections we’ve made in the manuscript. 

1. Age ranges from 42 to 75 years and can be a confounder. Please test age differences between the CAT and the SCT group (a standard T-test will do the job). Two are the possible outcomes. Either age means do not significantly differ between groups; in this case the randomization procedure did a good job and the authors just need to display the result of the t-test, to reassure the reader. Or one of the two groups is significantly older (or younger) than the other one; in this case the ANOVA analysis must be extended to a regression analysis, where age is included as a confounder.

No statistically significant differences between CAT and SCT groups in age were observed (mean age = 61.21 + 8.03 vs. 57.07 + 9.57, respectively; t statistic = -1.24; p = 0.22). This information was added to the results section. 

2. Page 9, Study participants section: "Fifty-eight patients ... 14 participants". As it often happens, there are missing values and drop-outs in this study. This is not necessarily a problem, provided that the missing values are ignorable. Missing values are ignorable when the probability of a missing value does not depend on the unobserved value. Do we have any information about the missing outcomes? If we can assume that the missing values are missing at random (i.e., neither older nor younger than the included subjects; neither in better nor in worse health status, and so on), then we can safely ignore them, as the authors did in their analysis. Otherwise, the results of the analysis could be possibly questionable.

Thank for introducing this very important issue. In order to test whether or not data are missing completely at random (i.e., p > 0.05), Little's test of missing completely at random test (Little, 1988) was conducted. This test is useful for testing the assumption of missing completely at random for multivariate, partially observed quantitative data (Li, 2013). The test's assumption is that the missingness of the data is independent of both the unobserved and the observed data (Graham, 2009). In the current study, Little's Missing data analysis showed that data were missing completely at random (Chi-square distance = 89.50, p = 0.32). This information was added to the statistical analysis section. 

Little RJA. A Test of Missing Completely at Random for Multivariate Data with Missing Values. Journal of the American Statistical Association. 1988 December;83:1198–1202.

Li C. Little’s Test of Missing Completely at Random. The Stata Journal. 2013 December;13:795–809.

Graham JW. Missing Data Analysis: Making It Work in the Real World. Annu Rev Psychol. 2009 January;60:549–576.

3. Normality. The whole analysis relies on an assumption of normally-distributed outcomes. Could the authors give some evidence of this assumption? If serious departures from normality are detected, then p-values could be biased.

Normality assumption was evaluated using the Shapiro-Wilk test. The analysis revealed the all study variables are normally distributed (p > 0.05) with W ranging from 0.90 to 0.94. This information was added to statistical analysis section. 

Shapiro SS, Wilk MB (1965) An analysis of variance test for normality (complete samples). Biometrika 52: 3-4.

4. Abstract, first line: "Exercise is a valuable intervention modality post-myocardial infarction (MI)." I think a "in" or a "for" is missing here: please check.

The sentence was corrected: Exercise is a valuable intervention modality for patients post-myocardial infarction (MI).

5. An obvious limitation of the study is that women are not considered. This should be remarked in the discussion, perhaps referring to other studies that include women.

Thank you for the comment. This study constrained to a small sample of patients, thus authors had to select more homogeneous group of male gender. Title of the manuscript was also rephrased to reflect the above: Intermittent aerobic-resistance interval training versus continues aerobic training: improvement in cardiac electrophysiologic and cardio-metabolic measures in male patients post myocadiac infarction.

In addition, as advised we have also referred to your important comment in the limitation section:

The current study was subject to several limitations. First, the generalizability of the results is in question owing to the single center and the small sample size of male only patients in each study group. The inclusion of only male patients is a common characteristic of numerous studies examining exercise training effects on heart failure patients. For example, in a recent meta-analysis in the subject it was reported that the majority of patients in cardiac rehabilitation and exercise training trials are males (77%) [46]. 

Tucker WJ, Beaudry RI, Liang Y, Clark AM, Tomczak CR, Nelson MD, Ellingsen O, Haykowsky MJ. Meta-analysis of Exercise Training on Left Ventricular Ejection Fraction in Heart Failure with Reduced Ejection Fraction: A 10-year Update. Progress in Cardiovascular Diseases. 2019 March;62:163–171.

6. Figure 3: "The vertical line extends from minimum to maximum values, excluding the outside values displayed as separate points". Please clarify how were outliers defined, as there is not a universal definition of outlying observations.

An outside value is defined as a value that is smaller than the lower quartile minus 1.5 times the interquartile range, or larger than the upper quartile plus 1.5 times the interquartile range; the middle line represents the median. This information was added to the figure's notes. 

7. Data availability. Honestly, I don't see specific reasons that would prevent the full availability of the data.

Data will now be available.

---

## [Decision Letter · Decision Letter 1]

5 Apr 2022

PONE-D-22-01160R1Intermittent aerobic-resistance interval training versus continues aerobic training: improvement in cardiac electrophysiologic and cardio-metabolic measures in male patients post myocadiac infarction, a randomized control trialPLOS ONE

Dear Dr. Barak,

Thank you for submitting your manuscript to PLOS ONE. After careful consideration, we feel that it has merit but does not fully meet PLOS ONE’s publication criteria as it currently stands. Therefore, we invite you to submit a revised version of the manuscript that addresses the points raised during the review process.

We look forward to receiving your revised manuscript.

Kind regards,

Walid Kamal Abdelbasset, Ph.D.

Academic Editor

PLOS ONE

Journal Requirements:

Reviewers' comments:

Reviewer's Responses to Questions

**Comments to the Author**

1. If the authors have adequately addressed your comments raised in a previous round of review and you feel that this manuscript is now acceptable for publication, you may indicate that here to bypass the “Comments to the Author” section, enter your conflict of interest statement in the “Confidential to Editor” section, and submit your "Accept" recommendation.

Reviewer #1: All comments have been addressed

Reviewer #2: (No Response)

Reviewer #3: All comments have been addressed

2. Is the manuscript technically sound, and do the data support the conclusions?

Reviewer #1: Yes

Reviewer #2: Yes

Reviewer #3: (No Response)

3. Has the statistical analysis been performed appropriately and rigorously? 

Reviewer #1: Yes

Reviewer #2: Yes

Reviewer #3: (No Response)

4. Have the authors made all data underlying the findings in their manuscript fully available?

Reviewer #1: Yes

Reviewer #2: No

Reviewer #3: (No Response)

5. Is the manuscript presented in an intelligible fashion and written in standard English?

Reviewer #1: Yes

Reviewer #2: (No Response)

Reviewer #3: (No Response)

6. Review Comments to the Author

Reviewer #1: Thanks for the important data presented in the manuscript. The revised data was done correctly. The research work exhibit great effort

Reviewer #2: Most of the comments have been addressed. few suggestions need further consideration and clarification.

- IN HRV, Correct Normalized power to normalized unit (nu)

_ Metabolic measures, I suggest changing it into "anthropometric measures" which is more common and describing the measures used

- It caught my attention one of the responses to reviewers that minimum EF was not mentioned, and the authors answered that inclusion criteria is based on NYHA. Does the author suggest that this classiffication is safer than including EF?

- Thanks for adding information about the medication and co-morbidities, but it would be better to include ut in a table with the common demographical data such as age, weight height, BMI

-modalities that may enhance the "intrinsic" cardiac recovery and peripheral metabolic health-related factors postMI (what does the author mean with intrinsic, if the author agree on "parasympathetic recovery" then I would use Extrinsic instead of intrinsic

Reviewer #3: (No Response)

7. PLOS authors have the option to publish the peer review history of their article (what does this mean?). If published, this will include your full peer review and any attached files.

Reviewer #1: No

Reviewer #2: No

Reviewer #3: No

---

## [Author Response · Author response to Decision Letter 1]

13 Apr 2022

Reviewer 1

Thank you very much for your feedback. We are happy that we were able to appropriately response to all your concerns. Your valuable comments helped improving the manuscript. 

Reviewer 2

1. IN HRV, Correct Normalized power to normalized unit (nu)

Corrected. 

2. Metabolic measures, I suggest changing it into "anthropometric measures" which is more common and describing the measures used.

Thank you very much for this comment. We changed in the entire manuscript "metabolic" to "anthropometric" except for places that cite from the literature references specifically related to metabolic measures.

3. It caught my attention one of the responses to reviewers that minimum EF was not mentioned, and the authors answered that inclusion criteria is based on NYHA. Does the author suggest that this classification is safer than including EF?

We used the NYHA classification as an inclusion criteria as it commonly used as a fundamental tool for risk stratification of heart failure and determines clinical trial eligibility and candidacy for drugs and devices. We did not try to suggest that this classification is safer than including EF. We rephrased the inclusion criteria to better explain the reason for using the NYHA. 

Caraballo C, Desai NR, Mulder H, Alhanti B, Wilson FP, Fiuzat M, Felker GM, Piña IL, O’Connor CM, Lindenfeld J, et al. Clinical Implications of the New York Heart Association Classification. JAHA. 2019 December 3;8:e014240.

4. Thanks for adding information about the medication and co-morbidities, but it would be better to include it in a table with the common demographical data such as age, weight height, BMI

We created a new table, table 1. The table has information regarding the 2 study's groups' demographic and clinical characteristics. We also added between-group analyses for both continues and categorical variables. We did not put in the table BMI as BMI is one of the study's outcome measures and it appears with the other outcome measures in tables 2 and 3. 

5. Modalities that may enhance the "intrinsic" cardiac recovery and peripheral metabolic health-related factors post MI - what does the author mean with intrinsic, if the author agree on "parasympathetic recovery" then I would use Extrinsic instead of intrinsic

Thank you for this comment. We reread the sentence and indeed it is not clear and accurate enough. We rephrased the sentence:

"Considering the effect of exercise on MI patients, it is vital to introduce novel training modalities that may enhance autonomic balance, intrinsic myocardial recovery and health related anthropometric factors post-MI."

Reviewer 3

Thank you very much for your time reviewing the manuscript. We are happy that we were able to appropriately response to all your concerns. Your valuable comments helped improving the manuscript.

---

## [Decision Letter · Decision Letter 2]

19 Apr 2022

Intermittent aerobic-resistance interval training versus continues aerobic training: improvement in cardiac electrophysiologic and anthropometric  measures in male patients post myocadiac infarction, a randomized control trial

PONE-D-22-01160R2

Dear Dr. Barak,

We’re pleased to inform you that your manuscript has been judged scientifically suitable for publication and will be formally accepted for publication once it meets all outstanding technical requirements.

Kind regards,

Walid Kamal Abdelbasset, Ph.D.

Academic Editor

PLOS ONE

Additional Editor Comments (optional):

Reviewers' comments:

Reviewer's Responses to Questions

**Comments to the Author**

1. If the authors have adequately addressed your comments raised in a previous round of review and you feel that this manuscript is now acceptable for publication, you may indicate that here to bypass the “Comments to the Author” section, enter your conflict of interest statement in the “Confidential to Editor” section, and submit your "Accept" recommendation.

Reviewer #2: All comments have been addressed

2. Is the manuscript technically sound, and do the data support the conclusions?

Reviewer #2: Yes

3. Has the statistical analysis been performed appropriately and rigorously? 

Reviewer #2: Yes

4. Have the authors made all data underlying the findings in their manuscript fully available?

Reviewer #2: Yes

5. Is the manuscript presented in an intelligible fashion and written in standard English?

Reviewer #2: Yes

6. Review Comments to the Author

Reviewer #2: The authors successfully amended the manuscript, and without a doubt it is a very informative piece of work.

7. PLOS authors have the option to publish the peer review history of their article (what does this mean?). If published, this will include your full peer review and any attached files.

Reviewer #2: No

---

## [Editor Report · Acceptance letter]

25 Apr 2022

PONE-D-22-01160R2 

Intermittent aerobic-resistance interval training versus continues aerobic training: improvement in cardiac electrophysiologic and anthropometric  measures in male patients post myocadiac infarction, a randomized control trial 

Dear Dr. Barak:

I'm pleased to inform you that your manuscript has been deemed suitable for publication in PLOS ONE. Congratulations! Your manuscript is now with our production department. 

Kind regards, 

on behalf of

Dr. Walid Kamal Abdelbasset 

Academic Editor

PLOS ONE